# The Association of Insomnia with Febrile Neutropenia, Leucopenia, and Infection in Women Receiving Adjuvant Chemotherapy for Breast Cancer

**DOI:** 10.3390/cancers17111838

**Published:** 2025-05-30

**Authors:** Audreylie Lemelin, Josée Savard, Michelle Chen, Lois E. Shepherd, Margot Burnell, Mark N. Levine, Bingshu E. Chen, Julie Lemieux

**Affiliations:** 1Hôtel-Dieu de Lévis, Lévis, QC G6V 3Z1, Canada; audreylie.lemelin.med@ssss.gouv.qc.ca; 2Cancer Research Centre, Faculty of Medicine, Université Laval, Québec, QC G1V 0A6, Canada; josee.savard@psy.ulaval.ca; 3School of Psychology, Université Laval, Québec, QC G1V 0A6, Canada; 4CHU de Québec, Université Laval, Québec, QC G1R 4G7, Canada; 5Canadian Cancer Trials Group, Queen’s University, Kingston, ON K7L 3N6, Canada; 14mc101@queensu.ca (M.C.); lshepherd@ctg.queensu.ca (L.E.S.); bechen@ctg.queensu.ca (B.E.C.); 6Saint John Regional Hospital, Saint John, NB E2L 4L2, Canada; margot.burnell@horizonnb.ca; 7Department of Oncology, McMaster University, Hamilton, ON L8V 5C2, Canada; mlevine@mcmaster.ca

**Keywords:** breast cancer, insomnia, chemotherapy, febrile neutropenia

## Abstract

Insomnia can affect the immune system. Cancer chemotherapy can also affect the immune system. This study evaluated whether there was an association between insomnia and immune dysfunction (neutropenia, leucopenia, and the occurrence of infections) in women treated with chemotherapy after breast cancer surgery. Insomnia was associated with febrile neutropenia, a serious consequence of immune dysfunction in patients receiving chemotherapy. This association remained significant after adjusting for confounders. In addition, chemotherapy dose reductions (often necessary to manage adverse effects) were more common in women with insomnia.

## 1. Introduction

Insomnia is a frequent complaint in cancer patients, with a prevalence of 20 to 59%, varying according to tumor type and treatment phase [1,2]. It is most frequent in women with breast cancer (42 to 69%) [3], which is hypothesized to be related in part to menopausal symptoms induced by chemotherapy and hormone therapy.

The psychoneuroimmunological model postulates that psychological factors such as stress, depression, and insomnia can impact the immune system and the risk of infection [4,5]. Experimental studies suggest that sleep deprivation alters the production of cytokines [6,7,8] via gene expression alterations [9], thus interfering with the activation of immunity. Sleep deprivation also appears to affect the distribution of leucocyte subsets [10,11] and to decrease the number and activity of natural killer cells [12,13,14]. The available evidence shows that, despite leading to a higher absolute number of neutrophils, sleep deprivation decreases the granulocytes’ capacity to phagocytose bacteria and to activate NADPH oxidase [10,15]. However, these studies may not generalize to clinical insomnia, a disorder characterized by dissatisfaction with sleep and sleep difficulties that occur despite having adequate opportunity to sleep, rather than reduced sleep duration per se. Clinical insomnia could be less severe than sleep deprivation.

The effect of naturalistic sleep difficulties on immune functioning and the infectious risk is less documented. A study of healthy individuals showed lower counts of CD4+, CD8+, and CD3+ T cells in patients with insomnia than in good sleepers [16]. Studies that have looked at the direct relationship between sleep difficulties and infections showed that insomnia was associated with a decreased vaccine response, both for influenza [17] and hepatitis B [18], in immunocompetent individuals. An experimental study, in which a rhinovirus was inoculated into healthy adults, found that patients with sleep difficulties were more likely to develop the infection [19]. The immune system and sleep homeostasis engage in a bidirectional regulatory relationship, where immune signals directly influence sleep architecture while sleep shapes immune competence through neuroendocrine and cellular mechanisms. This interplay is critical for maintaining physiological balance and optimizing host defense [20,21].

Given that insomnia appears to impair the immune system’s response and that it is two to three times more frequent in cancer patients [22], it is relevant to evaluate the clinical consequences of sleep difficulties in this population.

The aim of this study was to evaluate the association between insomnia and febrile neutropenia, leucopenia, and infections in patients treated with adjuvant chemotherapy for breast cancer, using the Canadian Cancer Trial Group (CCTG) data collected for the MA.21 trial [23]. The primary hypothesis was that, after having controlled for potential confounders, patients with insomnia would have a significantly greater prevalence of febrile neutropenia than those without insomnia.

## 2. Materials and Methods

### 2.1. Study Design

This is a secondary analysis of data from the CCTG MA.21 trial, a randomized clinical trial that compared three chemotherapy regimens (detailed below): CEF, EC/T dose-dense, and AC/T, in 2104 patients with node-positive or high-risk node-negative breast cancer [23].

### 2.2. Participants

Details about the methodology of the MA.21 trial can be found elsewhere [23]. Briefly, participants were women 60 years of age or younger who had undergone a complete resection of node-positive or high-risk node-negative breast cancer, defined as a tumor 1 cm and one or more of Grade 3, estrogen receptor-negative, or lymphovascular invasion. All patients who completed at least one quality of life questionnaire during the study period were included in this analysis. QOL questionnaires were the European Organization for Research and Treatment of Cancer Quality of Life Questionnaire C30 (EORTC QLQ-C30) [24].

### 2.3. Treatment Groups

Patients enrolled were randomized into one of three regimens of adjuvant chemotherapy. The first group received CEF for six cycles of 28 days, consisting of cyclophosphamide 75 mg/m^2^ PO on days 1–14, epirubicin 60 mg/m^2^ IV on days 1 and 8, and fluorouracil 500 mg/m^2^ IV on days 1 and 8. All patients in this arm received prophylactic antimicrobial therapy, consisting of cotrimoxazole or ciprofloxacin continuously, and patients were allowed to receive granulocyte colony-stimulating factor (G-CSF) if needed. The second group received EC/T dose-dense (epirubicin 120 mg/m^2^ IV day 1, cyclophosphamide 830 mg/m^2^ IV day 1 every 14 days for 6 cycles, followed by paclitaxel 175 mg/m^2^ IV every 21 days for 4 cycles). Use of G-CSF 5 µg/kg from day 2 to 13 and erythropoietin 40,000 UI s/c weekly was mandatory, but no antibiotic prophylaxis was administered. The last group was treated with AC/T (doxorubicin 60 mg/m^2^ IV day 1, cyclophosphamide 600 mg/m^2^ IV day 1, every 21 days for 4 cycles, followed by paclitaxel 175 mg/m^2^ IV every 21 days for 4 cycles). Neither G-CSF nor antibiotic prophylaxis was mandatory in this group, but it could be added if needed. Criteria for the use of G-CSF in each chemotherapy arm were detailed in the MA.21 protocol [23]. All participants received radiotherapy, starting between two and eight weeks following the end of chemotherapy. Patients with positive hormone receptors had to start tamoxifen within two weeks following the end of chemotherapy.

### 2.4. Sleep Measures

Patients had to fill out the EORTC QLQ-C30 at baseline, four times during the chemotherapy treatments, at the 9-month and 12-month follow-up, and annually thereafter for a total of five years. The EORTC QLQ-C30 is a self-administered questionnaire developed and validated by the EORTC Study Group [25]. This questionnaire contains one item assessing sleep difficulties (Question 11: Have you had trouble sleeping?), with answers ranging from 1 (“not at all”) to 4 (“very much”). Data from a previous randomized controlled trial conducted by a member of our team (JS) among 62 women with depressive symptoms [26] showed that a score of 3 (“quite a bit”) or greater was the best cut-off score to identify a clinical case of insomnia when compared to an Insomnia Severity Index score [27] of 11 or higher (unpublished data). This score was associated with a sensitivity of 90.7% and a specificity of 94.7%. It was, therefore, decided to use this cut-off to distinguish patients with or without insomnia (good sleepers). Patients were classified as having insomnia if their answers met this criterion at least once between the baseline and the last chemotherapy treatment. Otherwise, they were considered good sleepers.

### 2.5. Primary Outcome

The primary outcome was the occurrence of febrile neutropenia during all chemotherapy cycles, regardless of the chemotherapy arm. The definition of febrile neutropenia was based on the National Cancer Institute Common Toxicity Criteria (NCI CTC) Version 2.0 [28], as used in the MA.21 trial, which was: fever ≥38.5 °C of unknown origin without clinically or microbiologically detected infection, with an absolute neutrophil count (ANC) < 1.0 × 10^9^/L.

### 2.6. Secondary Outcomes

Secondary outcomes were the occurrences of leucopenia and infection, which were defined using the NCI CTC version 2.0 [28]. White blood counts were collected at each chemotherapy cycle (Day 1 and Day 8 for arm 1, Day 1 for arms 2 and 3). The worst-grade events were used to classify patients. In an unplanned analysis, chemotherapy dose reduction was investigated as a secondary outcome.

### 2.7. Statistical Analyses

Descriptive statistics were used to summarize the baseline patients’ characteristics. Univariate analyses exploring differences between insomnia patients and good sleepers on each outcome were conducted using either a Chi-square test or Fisher’s exact test. Multivariate analyses were conducted using a logistic regression model in order to adjust for a set of pre-defined confounding covariates, including the use of certain medications (G-CSF, antibiotic prophylaxis), demographics (age, race, menopausal status [post-menopausal/unknown vs. pre-menopausal]), performance status using the ECOG scale, cancer stage (T and N), chemotherapy groups and the presence of anxiety or depression (score of 70 or less on the EORTC QLQ-C30 emotional functioning domain; items 21–24) [29]. A chi-square test was used to assess potential interactions. A second multivariate analysis model was used, which included the same covariates except for the emotional functioning score, in order to isolate the effect of depression and anxiety on the results and because of the strong correlation between insomnia and these psychological symptoms. The level of significance was fixed at a two-sided *p*-value less than 0.05 for all analyses.

To investigate the dose-response relationship between insomnia and febrile neutropenia and related outcomes, we explored associations with the number of questionnaires (0 to 5) for which patients were classified as having “insomnia” and according to whether insomnia was present at baseline or not. A linear trend test was used to assess the dose-response effect.

### 2.8. Ethics and Funding

The study was approved by the local ethics review board at the CHU de Québec-Université Laval and the CCTG. No funding was provided by third parties for this study. Analyses were conducted free of charge by the CCTG central office.

## 3. Results

### 3.1. Participants’ Characteristics

Among the 2104 patients included in the MA.21 trial, 1731 completed at least one quality of life questionnaire (1720 completed the EORTC QLQ-C30 questionnaire) and were included in this study. Patients’ baseline characteristics are presented in Appendix A. There was no clinical difference between patients who completed QOL questionnaires and the entire population. The median age was 48 years. Almost 70% of patients were pre-menopausal. There were no significant differences between the chemotherapy regimens regarding baseline patient characteristics.

### 3.2. Insomnia and Quality of Life Data

There was no statistically significant difference in the global quality of life between chemotherapy groups, evaluated by the EORTC QLQ-C30 global score (Appendix A). The proportion of patients with insomnia (score ≥ 3 on the EORTC sleep problem item across time points) ranged from 22.1% to 24.7%, with no statistically significant difference between groups.

### 3.3. Immune Parameters Data

The rate of febrile neutropenia in the overall sample was 14.4%. The chemotherapy arm significantly influenced the occurrence of this outcome; in the CEF arm, 21.8% of patients experienced at least one febrile neutropenia episode, while 16.3% and 4.9% had at least one episode in the EC/T and AC/T arms, respectively (*p* < 0.0001). The chemotherapy arm was also associated with grade ≥1 leucopenia and infections, with a statistically significant lower rate of each of these outcomes in the AC/T arm as compared with the other two arms (Appendix A). The occurrence of chemotherapy delays was 68.4% in the CEF arm, 75.7% in the EC/T arm, and 67.1% in the AC/T arm (*p* = 0.003). Dose reductions were significantly more frequent in the CEF (37.7%) and EC/T (35.1%) arms as compared to the AC/T arm (6.5%) (*p* < 0.0001). The use of prophylactic antibiotics also differed significantly between groups (*p* < 0.0001), as anticipated, since it was mandatory for the CEF group and optional for the others. Finally, the use of G-CSF was significantly higher in the EC/T group (99.5%) and the CEF group (41.4%) than in the AC/T group (15.8%) (*p* < 0.0001), as expected.

### 3.4. Primary Outcome

Using the EORTC QLQ-C30 definition of insomnia, febrile neutropenia was more frequent in patients with insomnia, with 16.3% of patients presenting at least one febrile neutropenia episode versus 12.2% in good sleepers (*p* = 0.01 in the univariate analysis, Table 1). After adjusting for all potential confounders in the multivariate analysis, the relationship remained statistically significant with an odds ratio of 1.45 (95% CI 1.07–1.97, *p* = 0.02, Table 2). The multivariate analysis was then repeated with the same covariates, excluding the emotional functioning score. In this analysis, the odds ratio for febrile neutropenia was 1.42 and was significant (95% CI 1.06–1.92, *p* = 0.02, Table 2).

### 3.5. Secondary Outcomes: Leucopenia

The distribution of leucopenia was different in insomnia vs. good sleepers (*p* = 0.02 in univariate analysis, Table 1), with more patients with insomnia having grade 4 leucopenia. Moreover, the difference was not statistically significant when adjusting for confounders, with an odds ratio of 0.97 (95% CI 0.74–1.28, *p* = 0.82, Appendix A) in the initial multivariate analysis and of 0.95 (95% CI 0.73–1.24, *p* = 0.69, Appendix A) in the analysis excluding the emotional functioning score.

### 3.6. Secondary Outcomes: Infections

Infections occurred in 33.7% of patients with insomnia and 29.3% of good sleepers (*p* = 0.24 in univariate analysis, Table 1). The difference remained non-significant in the multivariate analyses, with odds ratios of 1.13 (95% CI 0.91–1.40, *p* = 0.27, Appendix A) and of 1.14 (95% CI 0.92–1.40, *p* = 0.23, Appendix A), respectively, in the two models.

### 3.7. Secondary Outcomes: Chemotherapy Dose Reductions

Finally, dose reductions were required for 30.6% of patients with insomnia, compared with 21.8% of good sleepers (*p* < 0.0001 in univariate analysis, Table 1). The multivariate analysis revealed a statistically significant difference between the two groups after adjusting for confounders, with an odds ratio of 1.67 (95% CI, 1.30–2.15; *p* < 0.0001; Appendix A). It remained significant even after emotional functioning was removed from the covariates, with an odds ratio of 1.67 (95% CI 1.31–2.13, *p* < 0.0001, Appendix A).

### 3.8. Secondary Outcomes: Dose-Response Analyses

There was a significant association between the number of questionnaires on which a patient scored positive for “insomnia” throughout the chemotherapy cycles and the rates of febrile neutropenia and chemotherapy dose reduction, suggesting a dose-response relationship between persistent insomnia and immune function. Similarly, stratified data according to whether insomnia was present at baseline or not showed a significant association with febrile neutropenia and chemotherapy dose reduction. These analyses are presented in Appendix A.

## 4. Discussion

This study evaluated the association between insomnia and febrile neutropenia in patients undergoing adjuvant chemotherapy for high-risk localized breast cancer as part of the MA.21 trial. Results showed that insomnia was associated with febrile neutropenia compared to good sleepers. In an unplanned analysis, it was found that insomnia was significantly associated with chemotherapy dose reductions, a relationship that remained statistically significant in both multivariate analyses. Our findings also suggest that the persistence of insomnia, as reflected by a greater number of EORTC questionnaires suggesting the presence of insomnia, was associated with febrile neutropenia and chemotherapy dose reductions.

Previous studies on cancer patients showed that patients with insomnia had lower neutrophil counts and experienced more infections compared with good sleepers [30]. Overall, the present study suggests that insomnia could be associated with febrile neutropenia in cancer patients on chemotherapy, which is consistent with the literature [2,30,31]. Its role appears to be independent of general emotional functioning despite the well-established bilateral relationship of anxiety and depression with sleep patterns and efficiency [32,33]. The psychoneuroimmunological literature found that both depression and anxiety may have an impact on immune functioning [34,35,36]. Interestingly, Ruel et al. showed that insomnia diagnosed with a clinical interview was associated with infections in women treated for cancer with chemotherapy; this effect remained significant after controlling for anxiety and depression [30]. Hoopes et al. reported detrimental associations between irregular sleep patterns and circulating immune cells in healthy young adults [31]. The immune system and sleep are engaged in a bidirectional relationship in which the immune system regulates sleep, and sleep plays a role in immune homeostasis [20,21]. Reciprocal feedback mechanisms play roles in these relationships and involve cytokines (like tumor necrosis factor α, interferon, and interleukin 6), reactive oxygen species, neurotransmitters, and nuclear factor-κB-driven inflammation [20,21].

Considering the added impact of chemotherapy on blood cells, the impact of insomnia could even be more important in cancer patients than in healthy young adults. It must be stressed that insomnia is not equivalent to sleep deprivation or sleep fragmentation and is a less severe condition than the latter two conditions [37]. The present study examined insomnia, which is defined as dissatisfaction with sleep and sleep difficulties occurring despite having adequate opportunity to sleep, while sleep deprivation is characterized by reduced sleep duration. Future studies, ideally prospective, should consider more precise sleep assessments, including sleep quality, duration, latency, and episodes of waking during sleep.

Chronic insomnia is associated with an array of negative consequences, including fatigue, psychological distress, cognitive impairments, increased medical consultations, and decreased quality of life [38,39]. Together, the findings of this study suggest that insomnia, particularly when persistent over the course of chemotherapy, could also have an influence on immune parameters and, ultimately, on chemotherapy tolerance. If these results are confirmed in future studies, they may have important clinical implications for patient management during chemotherapy. Indeed, the patients could be encouraged to maintain healthy lifestyle habits, including good sleep. Patients could also be screened for possible conditions, such as obstructive sleep apnea. Although the present study could not determine the direction of causality (i.e., whether insomnia contributes to leucopenia or vice versa), properly managing sleep could be a hypothesis-generating approach for future studies. It is unknown if non-pharmacological and pharmacological interventions to improve sleep could decrease negative, clinically relevant impacts on chemotherapy adverse events or dose reduction. It further emphasizes the need to better detect and treat these problems in clinical practice. Sleep difficulties can be detected using a specific sleep item added to the Edmonton Symptom Assessment System [40], a tool that is commonly used in cancer centers to screen for psychological distress. In terms of treatment, while hypnotic medications are by far the most commonly used treatment, cognitive-behavioral therapy for insomnia is considered the treatment of choice because of its more durable effect and lower risks [41]. Its efficacy in treating cancer-related insomnia is well-established [42,43].

Nevertheless, future studies should examine the association between insomnia and neutropenia and infections in patients with different types of cancer, different stages (localized or metastatic), and different treatments (e.g., chemotherapy, immunotherapy, and novel oral targeted therapies).

Our study has limitations. First, a single item from a cancer-specific quality of life questionnaire was used to assess insomnia rather than a detailed sleep evaluation. It is possible that the proportion of patients with insomnia (23.5%) was underestimated when compared to other studies, which found prevalence rates of insomnia between 42 and 69% in breast cancer patients [3]. This lower prevalence could also be explained by the fact that the study included a majority of women with non-metastatic disease who had not yet started hormonal therapy. It would be interesting to explore the same hypothesis in a large-scale study in which all patients would complete validated sleep questionnaires or interviews to have a more thorough and accurate evaluation of the role of insomnia. Another limitation is that, since we used existing data from the MA.21 trial, information on some potential confounders was not available, such as socio-economic status, tobacco use, and use of psychotropic medications.

In the MA.21 trial, the prevalence of insomnia varied from one time point to another [23]. A previous study in cancer patients undergoing curative treatment showed that insomnia at one time point was associated with having an infectious episode at the next time point [30]. It would have been interesting to look at the temporal relationship between insomnia and febrile neutropenia in our study, but since quality-of-life questionnaires were not completed at each chemotherapy cycle, the information would have been incomplete. An increasing number of questionnaires scoring “insomnia” was associated with febrile neutropenia and dose reduction, reinforcing the potential association between insomnia and these outcomes. However, it is not possible to rule out the possibility of a reverse relationship where immune changes would induce sleep impairments. Indeed, the available data could not be used to infer causality between insomnia and immune outcomes. Further studies are needed to establish a causal relationship between insomnia and immunity.

Another limitation is that, since we used existing data from the MA.21 trial, information on some potential confounders was not available, such as socio-economic status, tobacco use, and use of psychotropic medications. Finally, since the sample was exclusively composed of breast cancer patients undergoing adjuvant treatment, the results cannot be generalized to all cancer patients nor to all chemotherapy regimens. However, the results of previous studies did not suggest that the impact of insomnia on immunity varied according to tumor types [30,44].

The main strength of this study resides in its sample size, which is larger than most studies on this subject. The sample was also representative of the population of patients with high-risk localized breast cancer. Another strength is that the outcomes were rigorously measured throughout the study period, allowing the evaluation of a dose-response relationship.

## 5. Conclusions

We found that insomnia was associated with febrile neutropenia compared with good sleepers in women undergoing adjuvant chemotherapy for high-risk locoregional breast cancer. We also observed an association between insomnia and chemotherapy dose reductions in an unplanned analysis. This suggests that insomnia could alter the tolerance to chemotherapy, which could interfere with optimal treatment and possibly with the risk of recurrence and survival. Further research is needed to evaluate the contribution of insomnia to other complications of chemotherapy and to establish causality.

## Figures and Tables

**Table 1 cancers-17-01838-t001:** Univariate analyses of primary and secondary outcomes.

Outcomes	EORTC QLQ-C30 Q11	
	Insomnia	Good Sleepers
**Febrile neutropenia**		
No (grades 0–2)	781 (83.7%)	701 (87.8%)
Yes (grades 3–5)	152 (16.3%)	97 (12.2%)
*p*	0.01	
**Leucopenia**		
Grade 0	157 (16.8%)	141 (17.7%)
Grade 1	135 (14.5%)	138 (17.3%)
Grade 2	184 (19.7%)	186 (23.3%)
Grade 3	214 (22.9%)	173 (21.7%)
Grade 4	243 (26.1%)	160 (20.1%)
Grade 5	0 (0%)	0 (0%)
*p*	0.02	
**Infection**		
Grade 0	619 (66.4%)	564 (70.7%)
Grade 1	68 (7.3%)	62 (7.8%)
Grade 2	158 (16.9%)	110 (13.8%)
Grade 3	87 (9.3%)	61 (7.6%)
Grade 4	1 (0.1%)	1 (0.1%)
Grade 5	0 (0%)	0 (0%)
*p*	0.24	
**Chemotherapy delay**		
No	271 (29.1%)	239 (30.0%)
Yes	662 (70.9%)	559 (70.0%)
*p*	0.68	
**Chemotherapy dose reduction**		
No	648 (69.5%)	624 (78.2%)
Yes	285 (30.6%)	174 (21.8%)
*p*	<0.0001	

Legend: EORTC QLQ-C30: European Organization for Research and Treatment of Cancer Quality of Life Questionnaire C30, Definitions of grade based on NCI CTC version 2.0. All numbers were rounded to one decimal place.

**Table 2 cancers-17-01838-t002:** Multivariate analyses for febrile neutropenia.

Predictor	Multivariate Analysis, Including the EF Domain Score		Second Multivariate Analysis Excluding Emotional Functioning Score	
	Odds Ratio (95% CI)	*p*	Odds Ratio (95% CI)	*p*
Insomnia (Yes vs. No)	1.45 (1.07–1.97)	0.02	1.42 (1.06–1.92)	0.02
G-CSF (Yes vs. No)	4.72 (3.22–6.92)	<0.0001	4.71 (3.21–6.90)	<0.0001
Prophylactic antibiotics (Yes vs. No)	3.94 (2.52–6.18)	<0.0001	3.95 (2.52–6.19)	<0.0001
Age	1.00 (0.97–1.02)	0.81	1.00 (0.97–1.02)	0.83
Race (Aboriginal vs. Caucasian)	0.51 (0.07–3.99)	0.52	0.51 (0.07–3.98)	0.52
Race (Asian vs. Caucasian)	1.56 (0.68–3.60)	0.30	1.56 (0.68–3.61)	0.30
Race (Black vs. Caucasian)	0.65 (0.28–1.52)	0.32	0.65 (0.28–1.53)	0.32
Race (Unknown vs. Caucasian)	0.53 (0.15–1.92)	0.34	0.53 (0.15–1.92)	0.33
Treatment Arm (CEF vs. AC/T)	1.22 (0.69–2.15)	0.50	1.22 (0.69–2.16)	0.49
Treatment Arm (EC/T vs. AC/T)	1.13 (0.68–1.88)	0.63	1.14 (0.69–1.89)	0.61
Menopausal Status (Post vs. Pre)	1.57 (1.05–2.35)	0.03	1.57 (1.05–2.35)	0.03
Performance Status (1+ vs. 0)	0.75 (0.49–1.14)	0.18	0.75 (0.49–1.14)	0.17
N Stage (1 vs. 0)	1.08 (0.77–1.51)	0.65	1.08 (0.77–1.50)	0.66
N Stage (2 vs. 0)	0.86 (0.46–1.60)	0.63	0.86 (0.46–1.59)	0.62
T Stage (2 vs. 1)	1.23 (0.89–1.69)	0.21	1.23 (0.89–1.69)	0.21
T Stage (3+ vs. 1)	1.32 (0.80–2.18)	0.28	1.32 (0.80–2.18)	0.28
Emotional Functioning domain score	1.00 (0.99–1.01)	0.74	NA	NA

Legend: CEF: Cyclophosphamide + Epirubicin + Fluorouracil, EC/T: Epirubicin + Cyclophosphamide, followed by paclitaxel, AC/T: Doxorubicin + Cyclophosphamide, followed by Paclitaxel, G-CSF: Granulocyte colony-stimulating factor. All numbers were rounded to two decimals. Insomnia is defined using EORTC criteria (Q11 score ≥ 3).

## Data Availability

Data are available upon reasonable request to the Corresponding Author.

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
