# Peer review of "The Association of Insomnia with Febrile Neutropenia, Leucopenia, and Infection in Women Receiving Adjuvant Chemotherapy for Breast Cancer"

_cancers, 2025, doi:10.3390/cancers17111838_

Round 1

Reviewer 1 Report

Comments and Suggestions for Authors

The authors present appropriate rationale for their hypothesis and study design. The introduction gives a quick overview.  The authors pull from data regarding sleep deprivation and immune function and this is also looped in to the discussion. However, I think it is very important to emphasize that insomnia is not sleep deprivation. The authors should also consider including in the introduction how the immune system plays a role in the homeostatic drive of sleep, thus recognizing the bidirectional relationship.  Under Methods, the authors note the primary endpoint is the "risk" of febrile neutropenia.  Although one may measure risk within a population, it appears the methods are obtaining the prevalence of febrile neutropenic events.  A similar question is raised in regards to the risk vs prevalence of leukopenia and the need to change the chemotherapy.  If the authors are measuring prevalence then they should reword the subsequent sections.  This may seem to be a minor point, but the study is designed to find the association of the subject endorsing "trouble with sleep" and discrete clinical events.  Subjects may note that trouble with sleep can also indicate excessive sleepiness. The authors indicate that trouble with sleep translates to insomnia (supported by unpublished data) and the article may be strengthened by showing that preliminary data.  In the results section, the tables are very detailed.  Since the article is focused on sleep and immunity it may help the reader focus if Table 1 was in the supplementary material.  Also the authors may wish to consider a graph of the key elements comparing the primary endpoints in the insomnia and good sleep conditions for easy reference for the reader. 

In the discussion, I also suggest the authors reconsider the word risk to something more clear for the reader, as this is an association and prevalence of discrete events.  The word risk also implies some causation, when the mechanism may be bidirectional, as in those predisposed for these immune events are more likely to have sleep issues. Thus including a couple sentences on how the immune system influences sleep would be helpful for the reader.   The authors should also again highlight that insomnia is not sleep deprivation, much as sleep fragmentation is not the same as sleep deprivation, each has its own impact on the brain and body.     

Author Response

The authors present appropriate rationale for their hypothesis and study design. The introduction gives a quick overview.  The authors pull from data regarding sleep deprivation and immune function and this is also looped in to the discussion. However, I think it is very important to emphasize that insomnia is not sleep deprivation. The authors should also consider including in the introduction how the immune system plays a role in the homeostatic drive of sleep, thus recognizing the bidirectional relationship. 

            Response: We agree with the Reviewer that insomnia is not as severe as sleep deprivation. It is why the Introduction included a statement (that we expanded) to that effect: “However, these studies may not generalize to clinical insomnia, a disorder that is characterized by dissatisfaction with sleep and sleep difficulties occurring despite having adequate opportunity to sleep rather than reduced sleep duration per se. Clinical insomnia could be less severe than sleep deprivation.”

            We also added a statement in the Introduction: “The immune system and sleep homeostasis engage in a bidirectional regulatory relationship, where immune signals directly influence sleep architecture while sleep shapes immune competence through neuroendocrine and cellular mechanisms. This interplay is critical for maintaining physiological balance and optimizing host defense [1,2].”

Under Methods, the authors note the primary endpoint is the "risk" of febrile neutropenia. Although one may measure risk within a population, it appears the methods are obtaining the prevalence of febrile neutropenic events.  A similar question is raised in regards to the risk vs prevalence of leukopenia and the need to change the chemotherapy.  If the authors are measuring prevalence then they should reword the subsequent sections.  This may seem to be a minor point, but the study is designed to find the association of the subject endorsing "trouble with sleep" and discrete clinical events. 

            Response: We thank the Reviewer for the comments. We agree that we examined the association between the prevalence of insomnia and the prevalence or occurrence of neutropenia, leukopenia, and chemotherapy dose reduction. We revised the manuscript accordingly.

Subjects may note that trouble with sleep can also indicate excessive sleepiness. The authors indicate that trouble with sleep translates to insomnia (supported by unpublished data) and the article may be strengthened by showing that preliminary data. 

            Response: We thank the Reviewer for the comment. The sensitivity was 90.7% and the specificity 94.7%. This was added in the text.

In the results section, the tables are very detailed.  Since the article is focused on sleep and immunity it may help the reader focus if Table 1 was in the supplementary material.  Also the authors may wish to consider a graph of the key elements comparing the primary endpoints in the insomnia and good sleep conditions for easy reference for the reader. 

            Response: We thank the Reviewer. We moved Table 1 to the Supplementary Materials (as the new Table S3). The tables were renumbered accordingly. Regarding graphs, it is our opinion that tables are more valuable since they provide the exact values, while adding graphs would not enhance the manuscript much. Nevertheless, if it a crucial point for the Reviewer, we can provide them.

In the discussion, I also suggest the authors reconsider the word risk to something more clear for the reader, as this is an association and prevalence of discrete events. The word risk also implies some causation, when the mechanism may be bidirectional, as in those predisposed for these immune events are more likely to have sleep issues.  

             Response: We agree with the Reviewer. It was edited accordingly.

Thus including a couple sentences on how the immune system influences sleep would be helpful for the reader.   

             Response: We added in the Discussion that “The immune system and sleep are engaged in a bidirectional relationship in which the immune system regulate sleep and sleep plays a role in immune homeostasis [1,2]. Reciprocal feedback mechanisms are play roles in these relationships and involve cytokines (like tumor necrosis factor α, interferon, and interleukin 6), reactive oxygen species, neurotransmitters, and nuclear factor-κB-driven inflammation [1,2].”

The authors should also again highlight that insomnia is not sleep deprivation, much as sleep fragmentation is not the same as sleep deprivation, each has its own impact on the brain and body. 

             Response: We agree with the Reviewer. We added a statement: “It must be stressed that insomnia is not the equivalent to sleep deprivation or sleep fragmentation and is a less severe condition than the latter two conditions [3].”  

References

  1. Garbarino, S.; Lanteri, P.; Bragazzi, N.L.; Magnavita, N.; Scoditti, E. Role of sleep deprivation in immune-related disease risk and outcomes. Communications Biology 2021, 4, 1304, doi:10.1038/s42003-021-02825-4.
  2. Irwin, M.R.; Opp, M.R. Sleep Health: Reciprocal Regulation of Sleep and Innate Immunity. Neuropsychopharmacology 2017, 42, 129-155, doi:10.1038/npp.2016.148.
  3. Roth, T. Insomnia: Definition, Prevalence, Etiology, and Consequences. Journal of Clinical Sleep Medicine 2007, 3, S7-S10, doi:10.5664/jcsm.26929.

Reviewer 2 Report

Comments and Suggestions for Authors

The authors present an interesting and potentially publishable study entitled "The Association of Insomnia with Febrile Neutropenia, Leucopenia, and Infection in Women Receiving Adjuvant Chemotherapy for Breast Cancer." While the topic is relevant and the findings are noteworthy, the manuscript requires substantial revisions before it can be considered for publication. Please see detailed suggestions below:

Table 1: Please include p-values for individual group comparisons to clarify the statistical significance of the observed differences.

A comparative analysis of sociodemographic variables between the insomnia and good sleeper groups is missing and should be added. This would strengthen the interpretation of potential confounding factors.

The discussion is currently very limited and does not really discuss the findings. While authors provide a comprehensive limitation description (which is good), they fail to properly discuss their findings, meaning, possible source of difference, clinical significance and how those results can be compared to previously published reports.

The manuscript should address the directionality of the observed relationship. Is it that insomnia negatively affects chemotherapy tolerance, or do chemotherapy-related side effects contribute to the onset or worsening of insomnia?

Author Response

Table 1: Please include p-values for individual group comparisons to clarify the statistical significance of the observed differences.

             Response: We thank the Reviewer. In fact, the P-values for Table 1 are from chi-squared tests, which calculate a global P-value for the whole analysis. Please note that as per Reviewer #1’s request, Table 1 is now Table S3.

A comparative analysis of sociodemographic variables between the insomnia and good sleeper groups is missing and should be added. This would strengthen the interpretation of potential confounding factors.

             Response: We thank the Reviewer for this comment. However, we do not have access to socioeconomic data on this population such as income or education level that could be used as a surrogate for socioeconomic level. This was mentioned in the discussion: “Another limitation is that, since we used existing data from the MA.21 trial, information on some potential confounders was not available, such as socio-economic status, tobacco use, and use of psychotropic medications.”

The discussion is currently very limited and does not really discuss the findings. While authors provide a comprehensive limitation description (which is good), they fail to properly discuss their findings, meaning, possible source of difference, clinical significance and how those results can be compared to previously published reports.

            Response: We thank the Reviewer for the criticism about the Discussion. We revised the Discussion to try to improve it.

The manuscript should address the directionality of the observed relationship. Is it that insomnia negatively affects chemotherapy tolerance, or do chemotherapy-related side effects contribute to the onset or worsening of insomnia?

            Response: We agree with the Reviewer. Unfortunately, our data cannot be used to infer causality. It is was already noted in the Limitations, but we reinforced the statement.

Reviewer 3 Report

Comments and Suggestions for Authors

This is an interesting manuscript. However, detailed information is needed about HER-2 and the use of Herceptin with or without Perjeta (treatment that could cause leukopenia too). 

Author Response

This is an interesting manuscript. However, detailed information is needed about HER-2 and the use of Herceptin with or without Perjeta (treatment that could cause leukopenia too). 

            Response: The Reviewer raises a good point about the potential impact of trastuzumab and pertuzumab on blood cells. As indicated in the Methods, the treatments in the MA.21 trial were

  • CEF for six cycles of 28 days (Cyclophosphamide 75 mg/m2 po day 1-14, Epirubicin 60 mg/m2 IV day 1 and 8, Fluorouracil 500 mg/m2 IV day 1 and 8), or
  • EC/T dose dense (Epirubicin 120 mg/m2 IV day 1, Cyclophosphamide 830 mg/m2 IV day 1 every 14 days for 6 cycles, followed by Paclitaxel 175 mg/m2 IV every 21 days for 4 cycles), or
  • AC/T (Doxorubicin 60 mg/m2 IV day 1, Cyclophosphamide 600 mg/m2 IV day 1, every 21 days for 4 cycles, followed by Paclitaxel 175 mg/m2 IV every 21 days for 4 cycles).

Therefore, none of the patients included in the present study received an anti-HER2 therapy. Nevertheless, it was added as a future perspective.

Round 2

Reviewer 2 Report

Comments and Suggestions for Authors

I am satisfied with the changes in the discussion. Since no other sociodemographical data are available, then I have no further suggestions.